# Petrographic and Geotechnical Characteristics of Carbonate Aggregates from Poland and Their Correlation with the Design of Road Surface Structures

**DOI:** 10.3390/ma14082034

**Published:** 2021-04-18

**Authors:** Jerzy Trzciński, Emilia Wójcik, Mateusz Marszałek, Paweł Łukaszewski, Marek Krajewski, Stanisław Styk

**Affiliations:** 1Institute of Archaeology, Cardinal Stefan Wyszynski University, Wóycickiego 1/3, 01-938 Warsaw, Poland; 2Faculty of Geology, University of Warsaw, Żwirki i Wigury 93, 02-089 Warsaw, Poland; wojcike@uw.edu.pl (E.W.); mateusz.marszalek@yahoo.com (M.M.); pawel.lukaszewski@uw.edu.pl (P.Ł.); 3Eko Cinere Ltd., Warszawska 6/32, 15-063 Białystok, Poland; m.krajewski@ecocinere.pl (M.K.); s.styk@ecocinere.pl (S.S.)

**Keywords:** dolomite, limestone, rock structure, physical–mechanical properties, relationships, road engineering, asphalt concrete (AC), sustainability

## Abstract

The paper presents the basic problem related with practical application of carbonate rocks in construction: are carbonate aggregates produced from such rocks favorable for building engineering, particularly for road design and construction? To resolve this problem, (1) the geological-engineering properties of aggregates are presented, (2) the correlation between petrographic and engineering parameters is shown, and (3) a strict correlation between the geological-engineering properties and the freezing-thawing and crushing resistance is recognized. This knowledge has allowed to assess the usefulness of asphalt concrete (AC) made from dolomite and limestone aggregates in the design and construction of road surface structures. The petrography was characterized using optical microscopy and scanning electron microscopy (SEM) coupled with energy-dispersive X-ray spectroscope (EDS). Engineering properties were determined in accordance with European and Polish norms and guidelines. Statistical and design calculations were performed using dedicated software. The petrographic properties, and selected physical and mechanical parameters of the aggregates, were tested to show their influence on the freezing–thawing and crushing resistance. Strong functional relationships between the water adsorption, and the freezing–thawing and crushing resistance have been observed. Aggregate strength decreased after saturation with increasing concentrations of salt solutions. Calculations of AC fatigue durability and deformation allow for reducing the thickness of the road surface structure by about 20% in comparison to normative solutions. This conclusion has impact on the economy of road design and construction, and allows for a rational utilization of rock resources, which contributes to sustainable development of the construction industry.

## 1. Introduction

### 1.1. The Study in a Broad Context

Limestones and dolomites are carbonate rocks, commonly exposed on the surface of Earth’s lithosphere. As rock material, they have always been used in various branches of industry, particularly in construction. From the ancient times of Egypt’s Old Kingdom, when the beginnings of stone construction took place, such rocks were used to build monumental constructions, including the huge tombs of the pharaohs. These constructions are preserved today in a very good shape, which points not only to the skills of the ancient architects and craftspeople, but also to the good quality of this material.

Carbonate rocks are used worldwide in modern construction, including various types of aggregates applied in concrete and road constructions. However, the durability of concrete constructions with reference to the applied aggregate has not been sufficiently recognized and explained. There are very few complex reports on carbonate aggregates. Worth emphasizing are studies which discuss some issues on this topic from Europe, Asia, and North America [1,2,3,4,5,6,7,8,9,10,11,12,13,14,15,16]. The complex report on the application of limestones and dolomites in various branches of the industry in the USA, published by the Illinois State Geological Survey [17], draws particular attention. A similar case is with papers discussing the influence of environmental factors on aggregates, particularly the effect of low temperatures and salts. A good example of such study is the technical report published by the IBRI Icelandic Building Research Institute in cooperation with the SP Swedish National Testing and Research Institute [18].

### 1.2. Design of Road Surface Structures with Application of Carbonate Aggregates

In 2017–2021, with the financial support of the National Centre for Research and Development (NCBiR), a commercial enterprise conducted an innovative research–development project focused on designing, preparation and testing high stiffness ACs with application of different types of aggregates [19,20]. The essential part of this project involved comprehensive investigations of different types of aggregates, including carbonate aggregates. The performed studies allowed to prepare and conduct tests of commonly used and newly designed ACs. Selected results of these studies are presented herein.

In order to calculate the road surface structure, several original AC properties have to be determined. They include (1) contribution of free space in the mixture, (2) infilling of free space, (3) asphalt content, and (4) Poisson’s ratio. All these factors have direct impact on the stiffness ratio, which is the basic parameter influencing fatigue durability and thickness of road surface structures. The AC stiffness ratio is a specific property of the material resulting from the relationship of the stiffness ratio in different stress and strain states described by complex numbers.

Young’s modulus (E) is the ratio of linear strain or lateral elasticity. Its value indicates the elasticity of the material and is its specific feature. The modulus shows the relationship between relative linear strain *ε* of a material and stress σ, which occurs in the material in the range of elastic strain. This relationship is expressed by equation:σ/ε = E(1)

When a material displays rheological properties, as in the case of ACs, and load strain is delayed, then we refer to the stiffness module. This parameter is best described using complex numbers. The complex stiffness module E* is a complex number which can be described by equations and presented on a graph (Figure 1a):
E* = E′ + E″,(2)
E′ = │E*│ cos φ,(3)
E″ = │E*│ sin φ,(4)
where
E′—real part (elastic),E″—imaginary part (viscose).

Both components of the complex module are related with the phase angle according to the equation and presented in Figure 1b:
tg φ = E′/E″,(5)
where
φ—phase angle.

The stiffness module is the absolute value of the complex stiffness module. The value of the phase angle gives information of the prevalence of viscose or elastic properties in the material. Its lower value indicates a more elastic material. The phase angle is from 0° for steel to 90° for liquid. The phase angle results from the fact that in viscose-elastic bodies, strain is slightly delayed in relation to the applied load. The graphic interpretation of this phenomenon is presented in Figure 1b.

The criterion for assessing the viscose-elastic properties of ACs is the tangent of the phase angle; depending on the rheological properties of the material: (a) in viscose materials φ = 90°, then tg φ = ∞; (b) in elastic materials φ = 0°, then tg φ = 0; and (c) in viscose-elastic materials 0° < φ < 90°, then 0° < tg φ < ∞.

The ACs fatigue durability is their ability to contradict the destructive effect of short-term load related with traffic. It is an important factor influencing the exploitation time of a road surface structure. The value of the AC stiffness module influences fatigue durability and therefore also the thickness of the road surface structure.

### 1.3. The Main Aim of the Work

Since many years, the road and highway network in Poland is subject to intense development, favoring regional progress of the country and linking various parts of Europe. The transit routes from the south to the north and north-east and from the west to the east intersect in the country. Investments related with the construction of many kilometers of modern communication routes have increased the demand for high quality aggregates. In reference to these demands, in the frame of a MGśP project, PIG-PIB has elaborated and published data facilitating the search for usable rock deposits located close to the investment sites. All data for these areas are stored in a spatial database. So far, this is the first and only database for the entire area of Poland, allowing for determining rock material availability in a given area. The data are accessed on an internet platform and include the occurrence of a given rock deposit and information about its quality. This facilitates detailed recognition of the rock material with regard to selected sections of designed roads and highways, and covers an area of up to 20 km from the road axis [21].

References in engineering geology, geotechnics and geomechanics lack synthetic reports on different types of aggregates used in Poland, especially those of natural origin [22]. Stone materials, such as carbonate rocks—dolomites and limestones—occur in many parts of the country and have diverse applications [23]. There is general reluctance against using such rocks for construction, as they are considered less efficient than other materials, regardless their origin, age and lithology. Our goal is to fill this gap with regard to precise correlation of general geological knowledge, and the lithological and petrographic characteristics of carbonate rocks with the engineering properties of aggregates produced from these rocks. No strict correlations between the geotechnical properties of carbonate rocks and their resistance to environmental factors are known so far. Therefore, the main aims of this article include: (1) supplementation of geological–engineering knowledge on carbonate aggregates; (2) finding the correlation between the lithological and petrographic composition and the engineering properties of the rocks; and (3) finding a strict correlation between the geological–engineering properties, and the freezing–thawing and crushing resistance. These objectives are presented based on the geological-engineering characteristics of limestone and dolomite aggregates from two quarries located in Poland. Additionally, the correlation between the geotechnical properties and parameters characterizing the resistance to environmental factors was determined, and the usability of carbonate aggregates in road construction with regard to their application in mineral–asphalt mixtures was assessed. Innovative mixtures have been designed, which after testing gave promising results allowing for the reduction of the thickness of road surface structures and achieving a positive economic effect.

## 2. Materials and Methods

### 2.1. Materials

The studies were conducted on samples of two carbonate rocks, dolomite and limestone, acquired from two quarries belonging to the Lafarge-Holcim Group (Jona, Switzerland). The quarries are situated in Poland and their location is of strategic significance for the accessibility to such deposits in the northern and central parts of the country (Figure 2a).

The Radkowice quarry producing dolomite aggregate is located in south-eastern Poland in the Holy Cross Mountains (Figure 2a,c,e,g). In terms of geology and tectonics, the quarry lies in the western part of the Kielce Fold Zone, within the Gałęzice-Bolechowice Syncline [24]. The exploited material includes Middle and Upper Devonian dolomites. The beds dip at angles between 25° and 40° to the north. The thickness of Devonian strata reaches 600 m; locally, a several meter thick cover of Quaternary fluvioglacial sands occurs [25,26,27,28]. The lower part of the dolomites, attributed to the Eifelian, is represented by early diagenetic dolomites, whereas their upper part, representing the Givetian, is developed as epigenetic dolomites [29,30,31,32].

The production of dolomite aggregates in Radkowice dates back to 1946. At first the aggregates were made mostly for railroad construction, while at present the quarry produces crushed aggregates used in road construction, and for concrete and prefabricate production. The chemical composition of the exploited dolomites allows for their application in farming as calcium-magnesium fertilizers. The target production capacity of the quarry works is 2.5 mln Mg [33,34].

The Kujawy quarry is situated in central Poland (Figure 2a,b,d,f). With regard to geology and structure, the area lies in the axial part of the Mid-Polish Swell. In the Permian and Mesozoic, the area was located within the Mid-Polish Trough. Due to large subsidence, Permian, Triassic, Jurassic, and Cretaceous deposits in the area are characterized by large thicknesses. In addition to vertical tectonic movements, significant influence on the geological structure of the area had salt tectonics. The limestones exploited in Kujawy quarry developed during the Middle and Late Jurassic in a marine basin characterized by variable depth [35,36,37,38]. Limestone exploitation began already in the 19th century. At first, burnt and slaked lime was produced for the industry, construction and agriculture. At present, limestone blocks, two types of fine aggregate, eight types of coarse aggregates, an aggregate with continuous grain size, and a calcium fertilizer are produced. Annually, the quarry exploits about 4.8–5.2 mln Mg of limestones [33,34].

### 2.2. Methods

#### 2.2.1. Petrographic Analysis

The analyses concentrated on microscopic methods, which allow to perform investigations on small rock fragments and small surfaces of aggregate fragments. Thin sections were prepared from selected pieces of each rock type. The thin sections were analyzed using optical microscopy (Nikon Instruments Inc., Melville, NY, USA ) and scanning electron microscopy (SEM) (Jeol Ltd., Tokyo, Japan); additionally, fresh fractures were SEM viewed.

The thin sections were made on standard 26 × 48 mm microscopic slides. After cutting off from a larger fragment, a thin, ca. 3–5 mm thick sample was glued with resin to the slide and then polished with a successively finer abrasive powder to a thickness of about 0.02–0.03 mm (20–30 µm). The thin section surface was polished to a smoothness allowing for microscopic viewing and chemical analyses without coating.

Optical Microscopy

The thin sections were analyzed with Eclipe 100 (Nikon Instruments Inc., Melville, USA ) and E600 POL Nikon optical microscopes (Nikon Instruments Inc., Melville, USA ) equipped with digital cameras. The analyses included determination of rock texture and structure (arrangement of particular grain fractions and pore sizes, shape, rounding, and distribution of mineral components), mineral identification, as well as diagenetic changes and transformations, type of matrix, relics, and mineralization. Magnifications of up to 50 times were used.

Scanning Electron Microscopy (SEM);

The analyses were performed using a JSM-6380LA SEM (Jeol Ltd., Tokyo, Japan ) coupled with an EDS (Energy-dispersive X-ray spectroscope) Brucker GmbH XFlash 6/10 detector (Brucker, Berlin Germany ) (energy resolution with 126 eV at Mn Kα, 45 eV at C Kα, detection of elements from boron-B to americium-Am). The photographs were made using back scatter electron technology (BEI COMPO). Magnifications used were from 50 to 3500 times.

The performed analysis was aimed at determining the morphological properties and chemical composition of the grains and particles occurring in the analyzed samples. Thin sections and fresh fractures were used. The analyses were made in the following conditions: accelerating voltage—20 kV, beam current—70 mA, WD (working distance)—10 mm, duration of the chemical composition analysis—100 s. (live time). Analyses of elemental composition and viewing were made in the low vacuum mode (40 Pa). This technique does not require coating of the samples, therefore the chemical analyses are devoid of peaks from the element which is used to coat the samples in standard SEM analyses, i.e., carbon or gold. Analyses of the elemental composition were made in a non-standard mode and were normalized to 100% without including water bound in the OH groups. The obtained results should be treated as estimated values.

#### 2.2.2. Engineering Analyses

Geotechnical Parameters;

The following methods were used to determine the basic parameters of the dolomite and limestone aggregates (size 8/16): bulk density ρ_a_ [39], water adsorption WA_24_ [39], Los Angeles coefficient LA–resistance to fragmentation [40], resistance to crushing represented by the crushing strength in the dry mode X_r_, crushing strength after soaking X_m_, and decrease of strength X_s_ [41] linked with [42], and the freezing-thawing resistance F [43]. As the key parameter, water adsorption determines the values of other parameters. This value characterizes the ability of the aggregate to absorb and store water in pores and fractures. The parameter reflects the percentage loss of the mass of a water-soaked sample in relation to the sample mass after drying.

Resistance to Crushing after Saturation with Water and NaCl Solutions;

Preparation of carbonate aggregates was made according to [44] and the technical guidelines WT–1 [45], as well as [42], which has been withdrawn without being replaced by an actual norm [46]. Requirements from this norm were supplemented for the needs of the NCBiR project with guidelines from actual norms and additional processes of aggregate production. Additionally, the aggregates were saturated with water and NaCl solutions with 2% and 7% concentrations. After saturation, the aggregates were tested according to the modified procedure, and the results were compared with standard determinations in order to check the influence of water and NaCl on the strength of the aggregate. The analyses were performed for each type of aggregate, on 3 samples each collected from a large sample of the aggregate heaps. The carbonate aggregates were used for AC production.

#### 2.2.3. Examination of Asphalt Concrete (AC)

Stiffness Module;

The stiffness module was analyzed according to [47]. The test comprises cyclic bending of the sample in the shape of a bar inserted in the fatigue apparatus at a stable strain amplitude. Force, bar bending, phase angle, number of cycles, as well as extension stress and strain were registered during the tests. The stiffness module was calculated from the analyses. The following conditions of testing the complex stiffness module were adopted: temperature 10 °C, frequency 10 Hz, strain 50 mm/mm. The stiffness module was tested for AC HSM mixtures (asphalt concrete with high stiffness), designed in accordance with the Appendix to the regulation no. 54 of the Central Directorate of National Roads and Motorways from 18.11.2014 [48] treated as a national appendix to [49].

Fatigue Durability;

MWS Pavement Design software was used to calculate fatigue durability [50]. For calculating the construction state, i.e., stress, strain, and displacements, the program uses an analytical method based on the model of finite elastic layers. The surface is loaded with a vertical force, evenly distributed on a rectangular area. Full reciprocal incorporation occurs on the layer boundaries (displacement continuity), while settlement does not take place on the base of the lowermost layer. Thickness H_k_, Young’s modulus E and Poisson’s ratio ν_k_ are the parameters of each layer. Displacement, strain and stress values are calculated for the layer boundaries, with some of the strain and stress values being different above and below the layer boundaries, i.e., the discontinuities. The load model is presented in Figure 3.

The following assumptions were taken into account in the calculations—those linked with traffic load: traffic category—KR 7, number of acceptable calculation axles—53 mln axles 100 kN/lane, calculation time—20 years; and those linked with loading parameters: force—50.0 kN, contact pressure—0.85 MPa, load surface area—0.0589 m^2^, point load axis—X = 0, Y = 0. Road surface structures composed of mixtures with grain sizes up to 16 mm, 22 mm and 32 mm for dolomite; and 16 mm and 22 mm for limestone were compared. The reference structure was a susceptible road surface according to the Catalogue of Typical Structures of Susceptible Road Surfaces 2014 for traffic category KR 7. Values of geotechnical parameters of the ground substrate were attained according to group G1 carrying capacity, where the Young’s modulus E is 80 MPa and the Poisson’s ratio υ is 0.35. The carrying capacity group of the ground substrate classifies the carrying capacity of this substrate depending on the soil type and state, water conditions, swell properties, and characteristics of the road body. Four carrying capacity groups of the ground substrate have been distinguished: G1, G2, G3, and G4, according to the classification presented in the Catalogue of Typical Constructions of Susceptible and Semi-stiff Road Surfaces [51].

Two constructions have been compared in the calculations: (1) using AC HSM mixtures with thicknesses suggested in the catalogue [51] and (2) using AC HSM mixtures to ensure the required fatigue durability.

#### 2.2.4. Statistical Analysis

In order to analyze the influence of lithology, and selected physical and mechanical parameters of the aggregates on their freezing-thawing and crushing resistance, the results of investigations performed on 83 samples of dolomite aggregates and 75 samples of limestone aggregates, presented in Appendix A, have been used. The investigations were carried out in the IBMB laboratories (Building Materials Research Institute Ltd.) (IBMB, Warsaw, Poland) for Lafarge (at present Lafarge-Holcim). The following scheme of procedures has been applied for the correlation. In the first step, based on the entire dataset, correlation coefficients *R1* were determined between particular parameters and collated in correlation matrices. The Analysis Tool Pack for Microsoft Excel was used in this step. Due to extreme values of the parameters, 4.8% of the dolomite aggregate population and 1.3% of the limestone aggregate population was excluded from further analysis. After excluding extreme values, the values of the dependent variable (Y) were averaged for the same values of the independent variable (*X*); these data were used to construct plots. The plots indicate the trend lines of the linear function and the polynomial (best-fit) function. Values of the linear correlation coefficient before averaging (*R1*) and the best-fit value after averaging (*R2*) were determined. With regard to the correlation level, the correlation coefficient R is as follows: faint 0.0 < *R* ≤ 0.1, poor 0.1< *R* ≤ 0.3, average 0.3 < *R* ≤ 0.5, high 0.5 < *R* ≤ 0.7, very high 0.7 < *R* ≤ 0.9, almost full 0.9 < *R* < 1.0, full *R* = 1.

## 3. Results

### 3.1. Petrography

#### 3.1.1. Dolomite

On fresh fractures, the Radkowice dolomites are light to dark beige colored (Figure 4a). The rock is crystalline (dolosparite), the crystals shine on fresh fractures and the crystal size is identical, up to about 0.1 mm. Change of color to dark beige takes place on large irregular fragments up to several cm in diameter. Large calcite crystals up to 5 mm in size reflect the light within the matrix. Numerous calcite veins are up to 1 mm wide. In optical microscopy, dolomite structure and texture may be observed; they are characterized by the following features (Figure 4c,e):-mostly crystalline with variable crystal size;-crystals are mostly 10 to 30 µm in size, sometimes smaller or larger, but always above 4 µm, which points to sparite (dolosparite),-numerous fractures—veins filled with calcite (wider) and iron compounds (narrower), -relics of calcite crystals coated with developing dolomitization.

SEM analysis was aimed at determining the micro morphological features and chemical composition of the crystals in the dolomites. Based on that, transformations of carbonate rocks in the geological past could have been determined. The results are presented in Figure 4g,i. The rock is characterized by the following features which point to transformation processes:-dolomitization processes may be observed within individual dolomite crystals,-a lighter crust occurs around the crystals; it is not porous and does not contain white patches, considered as relics of the limestone rock, -the crystal center is clearly darker colored, with white patches and porous, -intercrystalline pores ranging in size from 10 to 20 µm are sporadic and intracrystalline pores ranging from 3 to 0.5 µm are more common in the center of the crystals; the pores are not in hydraulic contact with each other,-numerous regular and irregular caverns are filled with secondary mineralization, -fractures are common—veins filled with calcite mineralization, -lighter colored patches—limestone relics—occur in a darker dolomite matrix,-dolomitization reflected as autigenic dolomite crystals is visible against lighter colored calcite crystals. 

#### 3.1.2. Limestone

On fresh fractures the Kujawy limestones are light to dark grey in color (Figure 4b). The recognized fossils include numerous gastropod, bivalve and echinoderm fragments. The rock is a biomicrite, i.e., limestone composed of skeletal remains of organisms; in this case they include ooids, oncoids and zooids. Optical microscopy indicates that according to the Dunham classification [52], the rock is a packstone, while according to Folk [53,54] it is a biomicrite in which the allochems, typically represented by ooids and oncoids, are usually from 0.1 mm to 0.5 mm in size and comprise more than 50% of the rock. The space between the allochems is filled with a pelitic matrix and to a lesser degree, irregular calcite sparite (Figure 4d,f). According to microfacies classification, the analyzed rock is an ooid-oncoid packstone. Packstone is a limestone, whose depositional environment has not been clearly recognized [55]. A light grey matrix with small and large ooids and oncoids can be seen on the SEM images. The pore space consists of intercrystalline pores ranging from 50 to 150 micrometers and intracrystalline pores ranging from 2 to 15 micrometers. The pore space in the limestones partially shows hydraulic connectivity, especially in the intragranular pores. Chemical analysis of the matrix indicates that it is almost entirely composed of calcium carbonate (Figure 4h,j). Numerous dispersed and very fine pyrite grains have also been identified. 

### 3.2. Geotechnical Properties

The analyses in form of statistically condensed results are presented in Table 1 and Table 2 and Figure 5, Figure 6 and Figure 7. The analyzed parameters of dolomite and limestone aggregates are compared below. 

#### 3.2.1. Bulk Density (ρa)

Over 75% values for dolomite aggregate are within 2.80–2.84 Mg/m^3^, with the mean value 2.81 Mg/m^3^ (Table 1 and Figure 5a,b). For limestone aggregate, this parameter attains values between 2.70 and 2.73 Mg/m^3^ with the mean value 2.71 Mg/m^3^. According to aggregate classification with regard to ρ_a_ values, the analyzed dolomite and limestone aggregates may be considered as medium [56].

#### 3.2.2. Water Adsorption (WA_24_)

Values of *WA_24_* for dolomite aggregate distinctly differ compared to values for limestone aggregate. The values for dolomite aggregate are in a narrow range, while the results of *WA*_24_ for limestone aggregate attain a larger range (Table 1 and Figure 5c,d). *WA*_24_ values for dolomite aggregate show that over 90% of the results are within the dominant range of 0.7–1.2. For limestone aggregates, over 70% of the results are within the dominant range of 1.3–1.9. According to [39], the *WA*_24_ values classify the dolomite aggregate in category *WA*_24_ 1 (<1%) and *WA*_24_ 2 (≤2%), and the limestone aggregate in category *WA*_24_ 2, with results for 12 samples exceeding the upper limit for category *WA*_24_ 2 (see Appendix A).

#### 3.2.3. Los Angeles Coefficient (LA)-Resistance to Fragmentation

Histograms showing the distribution of *LA* values indicate that for both aggregates they attain different ranges (Figure 5e,f). Limestone aggregate has higher values of *LA* than dolomite aggregate (Table 1). Following [40], all analyzed samples, regardless their lithology, cannot be classified as high quality aggregates, because they are characterized by *LA* > 15. According to this norm, the *LA* values of dolomite aggregate classify the analyzed samples to categories LA20, LA25, and LA30, and the limestone aggregate samples—to categories LA30 and LA35. In the case of limestone aggregate, only a few samples attain extreme values, with *LA* over 30.

#### 3.2.4. Freezing–Thawing Resistance (F)

Values of *F* for dolomite aggregate are in a narrow range and over 56% of the results are within 0.4 and 0.8 (Figure 5g,h). Values of *F* for limestone aggregate are more variable, and the dominating range from 0.8 to 2.0 of the freezing–thawing resistance contains 73% of the results. The mean value of *F* for limestone aggregate is almost three times larger than for dolomite aggregate (Table 1). According to [43], the obtained results classify the dolomite aggregate to the freezing–thawing resistance categories F1 and F2. In turn, values of *F* for limestone aggregate classify this material to categories F1, F2, and F4, owing to the much larger values of this parameter. Only samples belonging to category *F1* according to [43] are classified as high quality aggregates, thus 75 out of 83 samples of dolomite aggregate and 12 out of 75 samples of limestone aggregate represent this category (see Appendix A).

#### 3.2.5. Crushing Strength (X_r_), Crushing Strength after Soaking (X_m_) and Decrease of Strength (X_s_)

Values of *X_r_* for dolomite and limestone aggregates oscillate in the same ranges, whereas values of *X*_m_ are distinctly different and in the case of limestone aggregate are characterized by larger variability (Table 1; Figure 6a–d). Comparison of the mean values of *X*_r_ and *X*_m_ points to larger differences in the values of these parameters in the case of *X*_m_ (6.4%). After soaking, dolomite aggregate is characterized by lower values of *X*_m_, thus has a higher crushing strength (Figure 6c). Decrease of strength *X*_s_ for dolomite aggregate attains a mean of 14.7%, which indicates the amount of strength increase after soaking. In the case of limestone aggregate, a positive value of *X_s_* was observed for 11 samples, which points to increase of strength after soaking (Figure 6d). For the remaining 64 samples, the decrease of strength *X*_s_ was negative, which indicates a lower strength of the limestone aggregate after soaking (Figure 6e,f). The mean decrease of strength for limestone aggregates was *X*_s_—12.8%, which suggests that they mostly have a lower strength after soaking. 

#### 3.2.6. Crushing Strength after Saturation with Water and NaCl Solutions

The results of aggregate analyses after saturation with water and NaCl solutions are presented in Table 2. They confirm the values of parameters presented in Table 1 and Figure 6 for water saturation. In turn, after saturation with NaCl solutions, dolomite and limestone aggregates also lose their strength, but to a lesser degree, and at higher concentrations of NaCl, the strength increases. Limestone aggregate gained more strength for higher NaCl concentrations.

### 3.3. Relationships

Table 3 presents the correlation coefficients with regard to lithological types of the analyzed aggregates. Data analysis indicates that for limestone aggregate these relationships have higher correlation coefficients than for dolomite aggregate, while deepened analysis linked with averaging the parameters of the aggregates significantly increased the correlation coefficient values (Table 3). For dolomite aggregate, only the relationship between bulk density and resistance to fragmentation is characterized by almost full correlation, which indicates that bulk density significantly influences resistance to fragmentation. In the case of the remaining relationships, poor, average, or lack of correlation was observed. For limestone aggregate, the correlation coefficients are very high or attain almost full correlation. Analysis of the relationship between water adsorption and the freezing–thawing resistance points to a strong link between these parameters in both lithological types.

Figure 7 presents the correlation relationships with the highest *R* for the entire dataset. The crucial parameter, best correlated with resistance to fragmentation and *F* value of the aggregates is *WA*_24_. Water adsorption analysis points to the larger water adsorption in the case of limestone aggregate. For the entire dataset, water adsorption varies in a wide range from 0.50 to 3.5%, with results for both aggregates between 1.0 and 1.5%. The analysis shows strong correlation between water adsorption and crushing and freezing–thawing resistance. The correlation relationships between water adsorption and crushing strength in dry *X*_r_ and saturated *X*_m_ conditions are reflected by a polynomial function, and the correlation coefficients are *R* = 0.67 and 0.86, respectively (Figure 7a,b). In the case of water adsorption *WA*_24_ and freezing–thawing resistance *F*, there is an almost full correlation in the analyzed dataset (Figure 7c). Increase of water adsorption causes decrease of crushing and freezing–thawing strength. Additionally, Figure 7d presents the influence of resistance to fragmentation *LA* on freezing-thawing resistance *F*. This relationship has a correlation coefficient of *R* = 0.83.

### 3.4. Asphalt Concrete (AC)

Results of the stiffness module analysis for ACs using limestone and dolomite aggregates are presented in Table 4 and Figure 8. Values of this parameter in the case of dolomite aggregate for each AC type are higher than for limestone aggregate. In turn, within this aggregate, slightly higher values of the stiffness module attain samples with a lower maximal grain-size, i.e., 16 mm, compared to those with a higher maximal grain-size, i.e., 22 mm.

The thickness of layers indicated in the Catalogue [51] and the stiffness modules obtained in the analyses for such materials allowed to receive much higher fatigue durabilities than those required for traffic category KR7. Table 5 compares the obtained values of fatigue durability for thicknesses according to the Catalogue [51] in relation to the specific calculations of fatigue durability and thicknesses ensuring indispensable parameters for traffic category KR7, i.e., 53,000,000 axels 100 kN/lane/20 years, obtained while using AC HSM mixtures. Application of such AC in each of the presented configurations allowed to decrease the thickness of the road structure by about 20%, i.e., from 3 to 5 cm in relation to the Catalogue solutions (Table 5; Figure 9).

## 4. Discussion

### 4.1. Petrography

The variability of carbonate rocks has resulted in the need to construct classifications of these rocks to enhance their recognition and nomenclature. The main classifications based on the percentage composition of the particular components were developed by Wiszniakow [57] and Pettijohn [58]. Both classifications refer to carbonate rocks, but each one includes different rock components. In the case of Wiszniakow (1956), the classification is based on two components: CaCO_3_ and CaMg[CO_3_]_2_, while Pettijohn’s (1975) classification is more precise, as it includes three components: CaMg[CO_3_]_2_, MgCO_3_ and MgO. Both classifications are based only on the chemical composition. This subdivision indicates that limestones pass evenly through dolomitic limestones and calcareous dolomites into dolomites.

The second, equally important classification of dolomite rocks is the classification based on their origin. Analysis of literature data suggests that the origin of dolomites took place in different conditions and at different diagenetic stages (see overview in [59]). According to most concepts, dolomite forms due to chemical transformation of calcium carbonate, related with the supply of the Mg^2+^ ion to the carbonate sediment at an early stage of diagenesis. Such process is known as dolomitization, and its course is reflected in the following formula: 2CaCO_3_ + Mg^2+^ = CaMg[CO_3_]_2_ + Ca^2+^. The process leads to the formation of secondary (metasomatic) dolomites. Primary dolomites, i.e., rocks formed due to direct precipitation of the mineral dolomite in specific marine and lake settings, are much less plausible. So far, the presence of a dolomite suspension, which would point to spontaneous nucleation of dolomite from sea water has not been observed in any natural aqueous environment (see overview in [59]).

In Poland, dolomites are common in numerous successions of different age, from the Precambrian to the Jurassic. Dolomite formations of economic significance occur in the Sudetes, Holy Cross Mountains and in the Silesia-Cracow region. They represent extremely important construction materials. 

The presence of clay minerals in limestones and dolomites used for the production of aggregate for roads and for the production of concrete plays an important role [60,61,62]. These minerals, especially those with a swelling structure, affect the quality of the aggregate and are responsible for damage to road and concrete structures. Petrographic analysis in SEM-EDS provided sufficient information about the presence of clay minerals in the studied rocks. The content of the elements Si, Al, Ca, Mg, and K, indicating the presence of these minerals, is negligible (Figure 4 g–j). Detailed analysis of the pore space of dolomites and limestones allowed to identify, on the basis of morphology and chemical composition, that a small content of clay minerals sporadically occurs in the corners of intercrystalline pores. Hydraulic contact between the pores is difficult, which protects the rocks against swelling under the influence of water solutions. Therefore, detailed studies on the content and type of clay minerals for these rocks are not required. The physical and mechanical parameters of the presented carbonate aggregates meet the standard requirements.

### 4.2. Geotechnical Properties

Analysis of bulk density *ρ*_a_ and other geotechnical parameters of dolomite and limestone aggregates indicates that their values are in the typical ranges for such rocks [15,63,64,65]. Values of *WA*_24_ for dolomite aggregate indicate that over 90% of the results are in the dominating range of 0.7–1.2% of water adsorption, which confirms the range of typical values (0.5–2.4%) for such aggregates [66]. Values of *WA*_24_ for limestone aggregate are within 1.1 and 3.9%, with a mean value of 1.7% (most samples show *WA*_24_ = 1.3–1.8%); therefore, they significantly exceed values typical for limestone aggregate, described in the literature [66]. Dolomite and limestone aggregates have typical values of *LA* as described in the literature [66]. In the case of limestone aggregate, a few samples attain extreme values, for *LA* exceeding 30, which is a consequence of local lithological variability of these rocks. Values of *X*_m_ of the analyzed aggregates are clearly variable, which is linked with the lithology and local petrographic variability of the rocks. Dolomite aggregate is characterized by typical *F* values for these rocks [15].

Increase of crushing strength after soaking (*X*_m_) is a specific feature of carbonate rocks in the case of metasomatic dolomites. In carbonate rocks, decrease of strength usually takes place after soaking. One of the main reasons of a different reaction of dolomites is their transformed origin, which caused the development of micro porosity, as observed in SEM analyses. During processes of transformation, micro porosity appeared in course of metasomatic dolomitization. The porosity results from the transformation of calcite crystals into dolomite crystals, which causes decrease of crystalline volume and increase of rock permeability, as confirmed by other studies (e.g., [67]). Pores with dimensions of several micrometers form a network, i.e., open (active) porosity. Carbonate rocks, particularly limestones, are characterized by closed (passive) porosity, i.e., individual pores do not communicate with each another. Therefore during soaking, the micro pores in dolomite aggregate became filled with water. In consequence, the process leads to increase of rock strength after soaking. The reason that caused increase of strength is water pressure in the micro capillaries with a concave meniscus. Water pressure in the capillaries assimilated part of the elastic strain, decreasing the susceptibility of the rock to destruction (crushing), which always takes place due to brittle strain. In consequence, it is recorded by higher values of *X*_m_ for dolomite aggregate.

Slight increase of the limestone aggregate strength after soaking with high concentration NaCl solutions is related with higher rock porosity. After saturation with NaCl solutions the calcareous rock becomes stronger due to crystallization of NaCl in its pores. Filling of free pore spaces by salt crystals during compression causes additional strength increase, because part of the exerted pressure is taken by the deformed structure of the crystallized salt. Decrease of strength was observed in dolomite aggregate, which is linked with deterioration of the pore space composed mainly of micro pores. This issue requires further detailed microscopic studies of porosity in order to explain the phenomena taking place during saturation with water and NaCl solutions. 

The performed studies do not allow for drawing final conclusions on the influence of water and salt saturation on the decrease of crushing strength in carbonate aggregate. It was assumed that the number of samples enables only to distinguish a trend but is too small to present final conclusions. Preliminary observations show that the observed trend has to be confirmed by studies on a larger number of samples. 

### 4.3. Correlation

Extreme values, much deviating from those typical for the analyzed population, have been excluded from the correlation. These extreme values are linked with local lithological variability, variable degree of rock weathering, and other natural factors. 

Limestone aggregate is characterized by high correlation coefficients in the analyzed relationships of *X*_r_, *X*_m_, and the freezing–thawing resistance *F* with water adsorption *WA*_24_ (Table 3). This is linked with the fact that water adsorption is strictly related with rock porosity. In the case of limestone aggregate, it is by about 1.9% higher than the porosity of dolomite aggregate, as confirmed by [64] and our studies. Water entering the pore space during freezing increases its volume and destroys the rock structure by micro crushing. This leads to its easier fragmentation under the influence of mechanical factors due to crack propagation along micro fractures, where crushing is facilitated. Due to this, strength parameters for limestone aggregate will be lower than those for dolomite aggregate. In addition to the obvious influence of porosity on the freezing–thawing resistance, this parameter has impact on resistance to crushing and fragmentation. This is because increasing volume of the pore space in a rock causes decrease of its load strength. The fact is also the result of crack propagation in places where micro pores occur. 

### 4.4. Asphalt Concrete (AC)

The research project presented in the article, particularly its part related to AC investigations, points to the change of fatigue durability depending on the stiffness modules studied herein. Calculations of fatigue durability indicate its significant increase for the accepted road surface structures. Structures adopted from the Catalogue of Typical Structures of Susceptible and Semi-stiff Surfaces, where stiffness modules of AC mixtures were replaced by stiffness modules of AC HSM mixtures, were compared with structures with a thickness of the bituminous layers reduced to ascertain fatigue durability for traffic category KR7. This procedure allowed for a ca. 20% reduction of the thickness of bituminous layers in road structures from the Catalogue. The hypothesis of a common possibility of reducing the thickness of particular layers in road structures would be very brave and premature, but the topic certainly requires further attention. An interesting issue would be the application of mixtures with a large stiffness module in lower rank road structures, e.g., KR 3-4. This might result in more common usage of mixtures which do not occur in Polish norms, such as Mixtures with discontinuous grain size (MNU) and SMA 16 (mastic-grit mixture with grain size up to 16 mm) representing JENA (Monolayer Asphalt Road Surface). This issue requires, however, further analysis, investigations, and calculations.

## 5. Conclusions

Based on the presented results of carbonate aggregate studies, it has been indicated that lithology and petrography are the key factors influencing their physical and mechanical properties.

The results reflect and evidence the complex interpretational approach to mineral materials such as natural aggregates. Appropriate aggregate selection may be of crucial significance for the strength parameters of road surfaces and construction elements exposed to the action of variable atmospheric conditions and subject to significant mechanical strains. 

Analyzes of dolomite aggregate from Radkowice quarry and limestone aggregate from Kujawy quarry allowed to draw the following conclusions:(1)Statistical analysis showed significant relationships between the physical and mechanical parameters and the resistance to crushing and freezing–thawing, which is particularly visible in the case of limestone aggregates.(2)The crushing strength of dolomite aggregates after saturation with salt solutions decreases due to deterioration processes.(3)The crushing strength of limestone aggregates after saturation with salt solutions increases as a result of salt crystallization.(4)Calculations of the durability of the road pavement showed its significant increase in the case of WMS mixtures.(5)The ACs fatigue durability and deformation calculations allow the thickness of the AC WMS pavement structure to be reduced by about 20%.

Detailed studies of rock material and aggregates produced from this material guarantee that the AC mixtures will be of very good quality. This in turn assures the production of a durable road surface with an optimal thickness, which would be resistant to environmental factors. Such approach among producers of aggregates, ACs, and road surfaces should assure a sustainable development and satisfaction of road users. Figure 10 presents the summary of the performed investigations.

## Figures and Tables

**Figure 1 materials-14-02034-f001:**
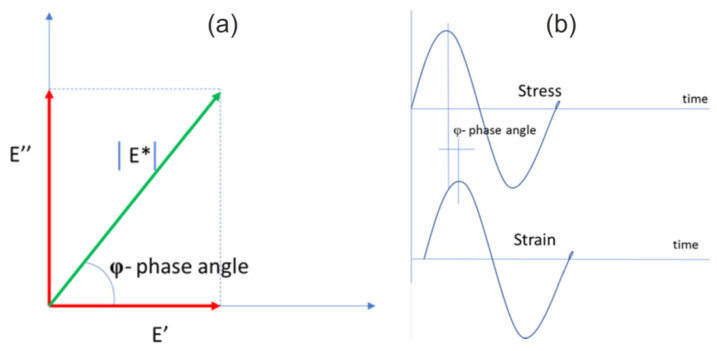
Complex stiffness module: (**a**) phase transfer angle; (**b**) delayed reaction of material strain in relation to loading.

**Figure 2 materials-14-02034-f002:**
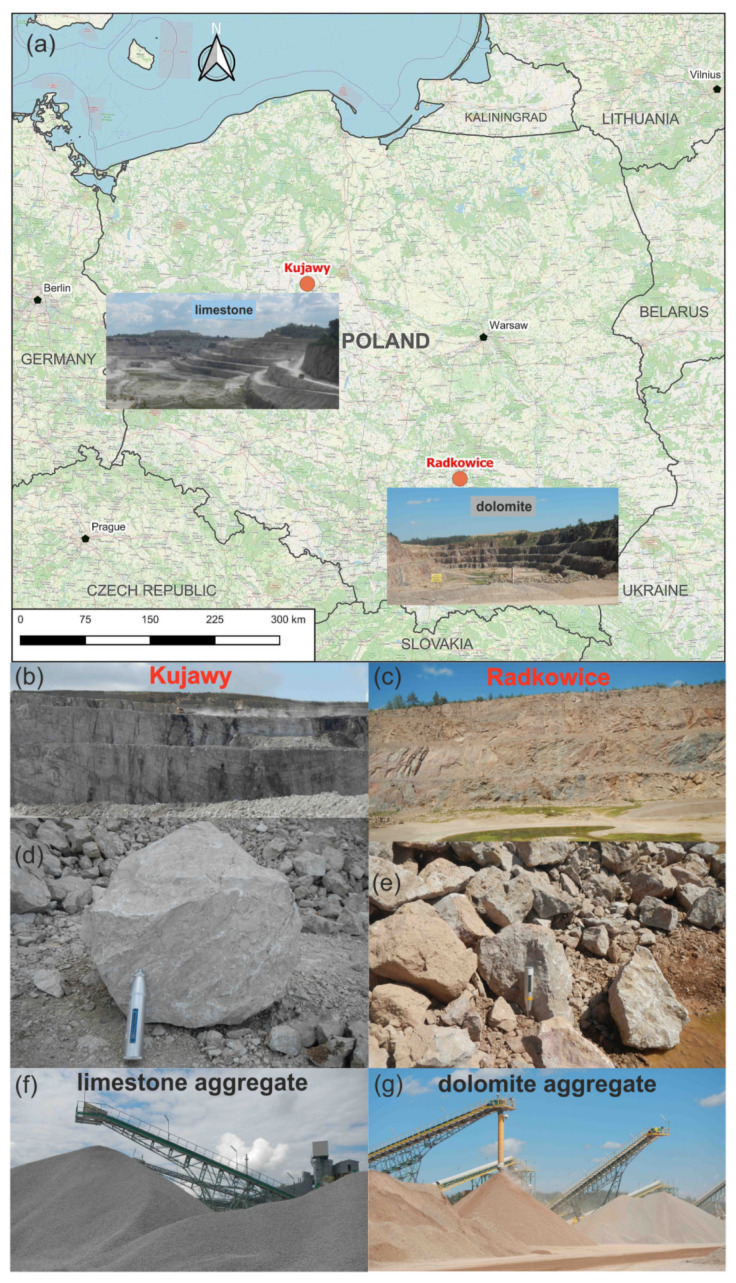
Radkowice dolomite quarry and Kujawy limestone quarry: location and general view of Kujawy quarry and Radkowice quarry (**a**); exploitation wall in Kujawy quarry (**b**) and Radkowice quarry (**c**); stone blocks in Kujawy quarry (**d**) and Radkowice quarry (**e**); aggregate heaps in Kujawy quarry (**f**) and Radkowice quarry (**g**).

**Figure 3 materials-14-02034-f003:**
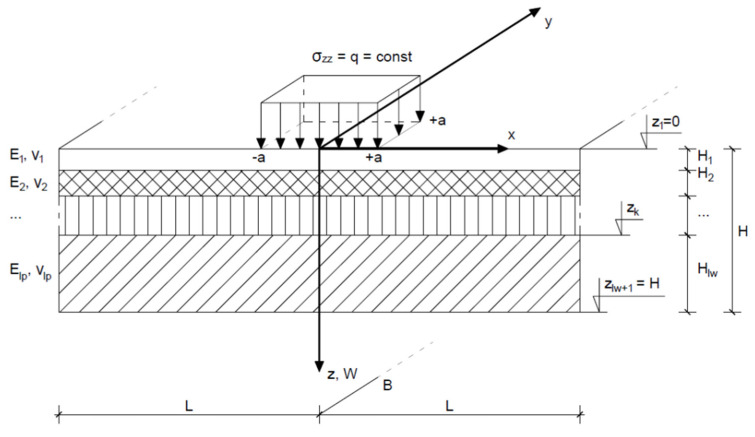
Loading model accepted in the calculations: ρ_zz_—normal stress; E_1_, E_2_—layer stiffness modules; υ_1_, υ_2_—Poisson’s coefficients; x, y, z—Cartesian coordinates; v, β, U, α—displacement and strain in different directions; H_1_, H_2_…H_n_—layer thickness; z_1_, z_2_, z_k_—depth and layer boundaries.

**Figure 4 materials-14-02034-f004:**
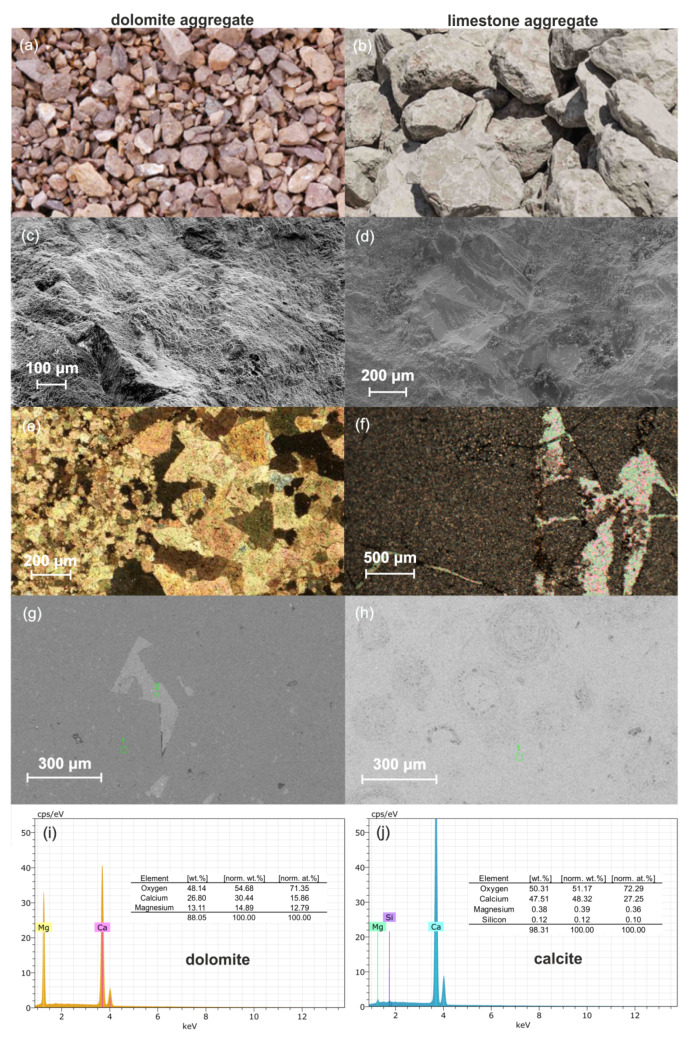
Petrography, mineralogy, structure and texture of dolomite and limestone aggregates: general view of dolomite (**a**) and limestone (**b**) aggregate; scanning electron microscopy (SEM) view of fresh fracture morphology of dolomite (**c**) and limestone (**d**) aggregate; rock structure in optical microscopy of dolomite (**e**) and limestone (**f**) aggregate (thin section in transmitted light, crossed nicols); SEM view of dolomite (**g**) and limestone (**h**) aggregate microstructure (thin section in BSE); EDS analysis–chemical and mineral composition of dolomite (**i**) and limestone (**j**) aggregate.

**Figure 5 materials-14-02034-f005:**
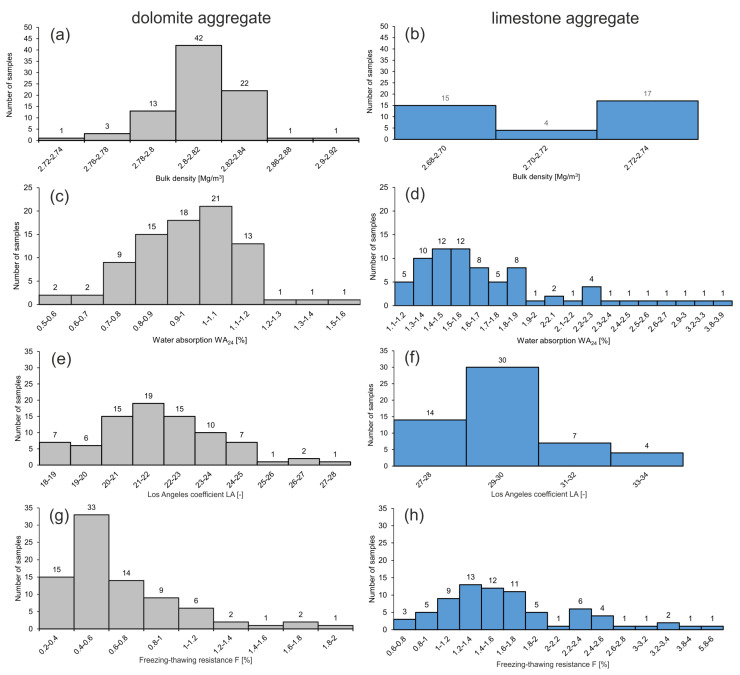
Distribution histograms of the values of geotechnical parameters for carbonate aggregate: bulk density of dolomite (**a**) and limestone (**b**) aggregate; water absorption WA24 of dolomite (**c**) and limestone (**d**) aggregate; Los Angeles coefficient LA–resistance to fragmentation of dolomite (**e**) and limestone (**f**) aggregate; (**d**) freezing–thawing resistance F of dolomite (**g**) and limestone (**h**) aggregate.

**Figure 6 materials-14-02034-f006:**
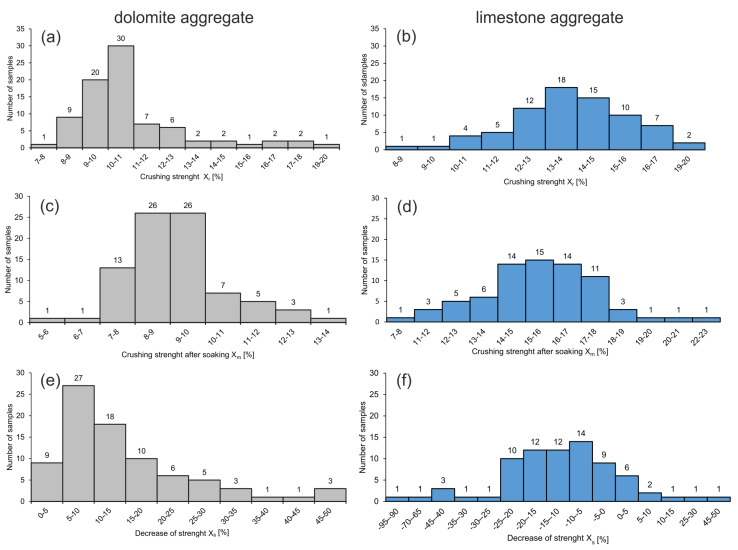
Distribution histograms of the values of geotechnical parameters for carbonate aggregates: crushing strength X_r_ of dolomite (**a**) and limestone (**b**) aggregate; crushing strength after soaking X_m_ of dolomite (**c**) and limestone (**d**) aggregate; decrease of strength X_s_ of dolomite (**e**) and limestone (**f**) aggregate.

**Figure 7 materials-14-02034-f007:**
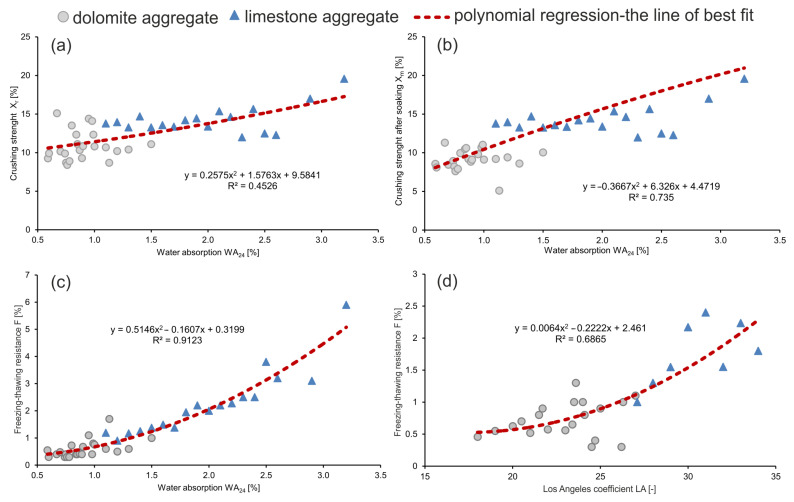
Relationships between geotechnical parameters for carbonate aggregates: (**a**) correlation between water absorption WA_24_ and crushing strength X_r_; (**b**) correlation between water absorption WA_24_ and crushing strength after soaking X_m_; (**c**) correlation between water absorption WA_24_ and freezing-thawing resistance F; (**d**) correlation between Los Angeles coefficient LA–resistance to fragmentation and freezing–thawing resistance F.

**Figure 8 materials-14-02034-f008:**
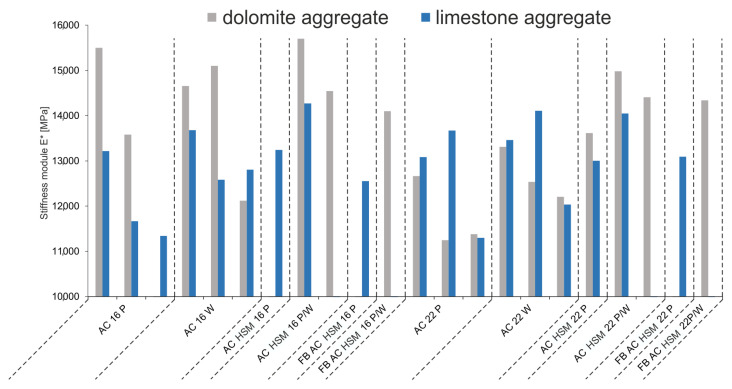
Values of stiffness module E* with regard to type of asphalt concrete (AC).

**Figure 9 materials-14-02034-f009:**
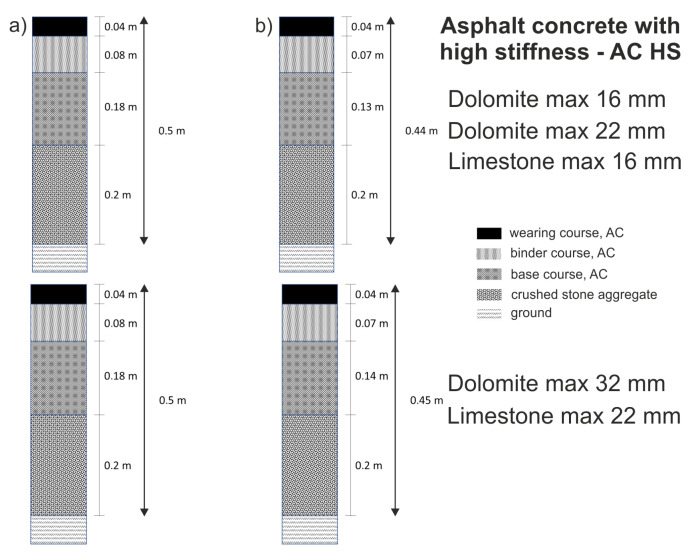
Construction of road structure KR 7 for various types of asphalt concrete (AC): (**a**) according to the GDDKiA Catalogue; (**b**) according to the studies presented herein (compare with Table 5). 16, 22, and 32—upper limit of aggregate in millimeters, lower limit is 0 mm.

**Figure 10 materials-14-02034-f010:**
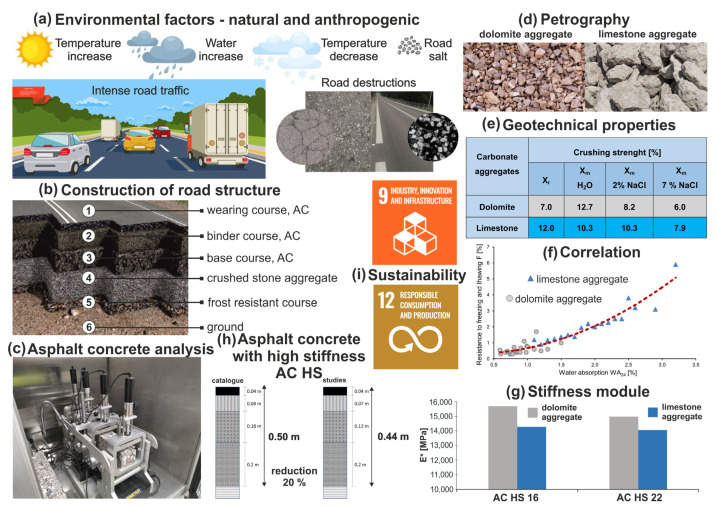
Summarized sustainability research on dolomite and limestone aggregates(**a**) Environmental factors, (**b**) Construction of road structure, (**c**) Asphalt concrete analysis, (**d**) Petrography, (**e**) Geotechnical properties, (**f**) Correlation, (**g**) Stiffness module, (**h**) Asphalt concrete with high stiffness—AC HS, (**i**) Sustainability.

**Table 1 materials-14-02034-t001:** Geotechnical parameters of dolomite and limestone aggregates. Statistic data: min—minimal value; max—maximum value; mean—average value; sd—standard deviation.

Parameter		Dolomite Aggregate	Limestone Aggregate
Bulk density ρ_a_ (Mg/m^3^)	min	2.73	2.70
	max	2.92	2.73
	mean	2.81	2.71
	sd	0.02	0.01
Water adsorption WA_24_ (%)	min	0.59	1.10
	max	1.50	3.90
	mean	0.93	1.68
	sd	0.15	0.48
Los Angeles coefficient LA (-)	min	18.00	27.00
	max	27.10	34.00
	mean	21.42	29.60
	sd	2.00	1.54
Freeze-thaw resistance F (%)	min	0.20	0.60
	max	2.00	5.90
	mean	0.62	1.66
	sd	0.35	0.80
Crushing strength X_r_ (%)	min	7.90	8.95
	max	19.35	19.96
	mean	10.86	13.88
	sd	2.14	1.93
Crushing strength after soaking X_m_ (%)	min	5.10	7.97
	max	13.10	22.45
	mean	9.13	15.49
	sd	1.32	2.16
Decrease of strength X_s_ (%)	min	1.09	−91.28
	max	49.29	45.71
	mean	14.67	−12.81
	sd	10.57	17.26

**Table 2 materials-14-02034-t002:** Geotechnical parameters of dolomite and limestone aggregates. Crushing strength before and after water and NaCl adsorption.

Carbonate Aggregate	Lower Size/Upper Size	Crushing Strenght (%)			
		X_r_	X_m_ after Soaking in H_2_O	X_m NaCl2_ after Soaking in 2% NaCl	X_m NaCl7_ after Soaking in 7% NaCl
Dolomite	4/8	6.1	16.4	13.5	11.2
	8/16	7.0	12.7	8.2	6.0
	16/22	12.8	14.3	12.4	11.4
Limestone	4/8	7.8	8.6	8.6	7.1
	8/16	12.0	10.3	10.3	7.9
	16/22	16.1	11.5	11.7	11.2

**Table 3 materials-14-02034-t003:** Correlation coefficient for different relationships: R1—data before averaging, linear function; R2—data after averaging, best-fit function.

Correlations	Correlation Coefficient			
	Dolomite		Limestone	
	R1	R2	R1	R2
X_r_/ρ_a_	−0.35	0.91	−0.14	−0.66
X_r_/LA	0.37	0.35	0.19	0.94
X_r_/WA_24_	0.004	0.06	0.23	0.71
X_m_/WA_24_	0	0.06	0.41	0.81
F/WA_24_	0.27	0.54	0.78	0.94
F/LA	0.30	0.24	0.30	0.80

**Table 4 materials-14-02034-t004:** Stiffness module depending on the type of asphalt concrete (AC). Explanation of symbols used for mixtures: AC—asphalt concrete; AC HSM—asphalt concrete with high stiffness; FB AC HSM—asphalt concrete with high stiffness with application of foam bitumen; P—foundation layer mixture; W—binding layer mixture; P/W—foundation or binding layer mixture; 16 and 22—upper limit of aggregate in millimeters, lower limit is 0 mm.

Type of Asphalt Concrete	Stiffness Module E* (MPa)	
	Dolomite Aggregate	Limestone Aggregate
AC 16 P	15,499	13,219
	13,580	11,665
	-	11,341
AC 16 W	14,656	13,679
	15,101	12,581
	12,119	12,804
AC HSM 16 P	-	13,242
AC HSM 16 P/W	15,701	14,270
	14,546	-
FB AC HSM 16 P	-	12,553
FB AC HSM 16 P/W	14,101	-
AC 22 P	12,665	13,084
	11,247	13,671
	11,381	11,297
AC 22 W	13,311	13,462
	12,537	14,109
	12,208	12,033
AC HSM 22 P	13,616	13,002
AC HSM 22 P/W	14,982	14,047
	14,407	-
FB AC HSM 22 P	-	13,091
FB AC HSM 22P/W	14,340	-

**Table 5 materials-14-02034-t005:** Fatigue durability and thickness of road surface structure: ^1^—according to GDDKiA Catalogue; ^2^—according to the studies presented herein (see Figure 9).

Carbonate Aggregate	Type of Asphalt Concrete	Fatigue Durability (Axel 100 kN/Line/20 Years)	Thickness of Road Structure (cm)
Dolomite	AC HSM 16	435,773,372 ^1^	50 ^1^
		76,602,819 ^2^	44 ^2^
	AC HSM 22	391,186,346	50
		66,071,240	44
	AC HSM 32	289,556,405	50
		69,621,783	45
Limestone	AC HSM 16	350,090,065	50
		59,983,402	44
	AC HSM 22	217,260,948	50
		59,791,915	45

## Data Availability

Eko Cinere Ltd., Warszawska 6/32, 15-063 Białystok.

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
