# Peer review of "Petrographic and Geotechnical Characteristics of Carbonate Aggregates from Poland and Their Correlation with the Design of Road Surface Structures"

_materials, 2021, doi:10.3390/ma14082034_

Round 1

Reviewer 1 Report

The manuscript: infrastructures-1166631 titled “Petrographic and Geotechnical Characteristics of Carbonate Aggregates from Poland and their Correlation with the Design of Road Surface Structures”, presents an interesting review study. The manuscript is well written, with Figures and Table being correctly displayed and informative. Authors clearly set the problem regarding the applicability of using carbonate aggregates for road design and construction by successfully combining the petrographic and engineering parameters to resolve this. Conclusions and Abstract are balanced, well written, and successfully display the major outlines of this paper. Therefore, I suggest that this paper should be published in its present form in the Journal of “Materials”.

Reviewer 2 Report

I am glad to read a very interesting and one of high quality study, which is well written. I think that after minor revisions it should be published in the journal.

  • A complete petrographic study of rocks which are used as aggregates in various applications and more specifically in construction applications includes except of the investigation under polarizing microscope the X-Ray Diffractometry (XRPD). In the case of rocks (mainly of sedimentary such as limestones and dolomites) which will be used as concrete aggregates, the identification of the type of clay minerals contained in those rocks seems to be crucial. Especially, it is crucial to identify if there are swelling clay minerals or not in the tested rocks as you know swelling clay minerals are responsible for failures in concrete constructions. For all the reasons mentioned above, I suggest to examine the clay fraction of the tested rocks in order to identify the presence of swelling clay minerals or not in the studied rocks.
  • As for the conclusions, I suggest to rewrite them more clearly in order to highlight your great work.
